# Chromium Nanoparticles Together with a Switch Away from High-Fat/Low-Fiber Dietary Habits Enhances the Pro-Healthy Regulation of Liver Lipid Metabolism and Inflammation in Obese Rats

**DOI:** 10.3390/ijms24032940

**Published:** 2023-02-02

**Authors:** Bartosz Fotschki, Katarzyna Ognik, Joanna Fotschki, Dorota Napiórkowska, Ewelina Cholewińska, Magdalena Krauze, Jerzy Juśkiewicz

**Affiliations:** 1Institute of Animal Reproduction and Food Research, Polish Academy of Sciences, 10 Tuwima Street, 10-718 Olsztyn, Poland; 2Department of Biochemistry and Toxicology, Faculty of Animal Sciences and Bioeconomy, University of Life Sciences in Lublin, 13 Akademicka Street, 20-950 Lublin, Poland

**Keywords:** nanoparticles, oxidative stress, inflammation, fatty liver diseases, obesity, rats

## Abstract

The study on Wistar rats was conducted to investigate the effects of a pharmacologically relevant dose 0.3 mg/kg body weight of chromium supplementation (commonly used picolinate or novel form as nanoparticles) and switching away from obesogenic dietary habits on the parameters of lipid metabolism, inflammation, and oxidative stress in liver and plasma. Favorable effects related to dietary changes from the obesogenic diet were considerably enhanced when the diet was supplemented with chromium nanoparticles. This combination exerted the strongest fat content and cholesterol reduction in the liver. Moreover, in this group, a favorable antioxidative effect was observed through GSH/GSSG elevation in the liver as well as ALT activity reduction in the plasma and IL-6 levels in the liver. The molecular mechanisms associated with regulating lipid metabolism, oxidative stress and inflammation might be related to lower expression of HIF-1α, COX-2, and LOX-1 and upregulation of PPARα in the liver. Supplementation with chromium nanoparticles without changes in the obesogenic diet also favorably affected lipid metabolism and oxidative stress in the liver; however, the examined effects were moderate. In conclusion, the favorable effects of switching from an obesogenic to a balanced diet on hepatic lipid metabolism, oxidative stress, and inflammation induced by an obesogenic diet might be enhanced by supplementation with chromium nanoparticles.

## 1. Introduction

Currently, overweight or obesity affects approximately 603 million adults worldwide and contributes to 4 million deaths annually [1]. To alleviate symptoms associated with an obese status, individuals are often prescribed weight loss. The patients increase physical activity, change dietary habits, and often decide to enhance the health-promoting effects with dietary supplements. However, a high-fat diet is important in the pathophysiology of obesity [1]. In recent decades, the use of Cr in human and animal nutrition has attracted the attention of researchers [2]. Chromium (III) (Cr) is a key microelement involved in the metabolism of carbohydrates, proteins, and fats in humans and animals. Reports have also indicated that chromium(III) is involved in the metabolism of nucleic acids and favorably stimulates the immune response and disease resistance [3]. Because of these properties of chromium, particularly its ability to regulate carbohydrate-lipid metabolism and reduce body weight, it is popularly used as a factor supporting the treatment of type 2 diabetes and as a component of supplements used in slimming (anti-obesity) treatments. However, the essentiality of chromium (Cr) has been questioned in some recent studies [4]. Although some studies have suggested that chromium supplementation decreases insulin levels, improves glucose disposal rates and lipid profiles, and beneficially reduces body weight in obese individuals [5,6], in other studies, Cr supplements to diabetic or healthy subjects did not indicate beneficial effects on glucose metabolism and diabetes [7]. In contrast to certain clinical human observations, studies with rodent models supplemented with Cr have unambiguously indicated certain roles of Cr as a pharmacologically active element in glucose tolerance [4]. In this respect, supplementing the diet of obese and/or normal-weight rats with the most popular Cr form, chromium picolinate (Cr-Pic), has been shown to decrease plasma malondialdehyde, glucose, insulin, total cholesterol, and triacylglycerol concentrations as well as improve glucose disposal rates and elevate GLUT-2 and GLUT-4 expression in the liver and muscle [8,9]. Sreejayan et al. [10] showed that markers of insulin resistance (phospho-c-Jun and IRS-1 phosphoserine) and hepatic ER stress (p-PERK, p-IRE-1, p-eIF2α), which were elevated in ob/ob mice, were attenuated following Cr treatment. The antioxidant and anti-inflammatory activities of Cr-Pic are mediated through the Nrf2 and NFκβ pathways by increasing and decreasing their activities, respectively [8,11]. Other factors proposed to be involved in hepatic recovery in obese individuals are PPARα (peroxisome proliferator-activated receptor alpha), COX-2 (cyclooxygenase 2), HIF-1α (hypoxia-inducible factor 1 alpha), LOX-1 (lectin-like oxidized low-density lipoprotein (LDL) receptor-1), and others [12].

However, because of the relatively low bioavailability of chromium picolinate, other forms of this element are sought that could be better utilized by the body. Thus, researchers are increasingly interested in complexes of Cr with amino acids as well as inorganic chromium nanoparticles (NPs) [3,13]. NPs tend to exhibit different properties than larger particles of the same element. After entering circulation, NPs interact with the endothelium and induce nitric oxide (NO) signaling impairment. Oxidative stress and the inflammatory response are the mechanisms of metal NPs [14]. Scientists agree that few studies have investigated the physiological and toxicological impact of nanomaterials, particularly in areas relating to the effects and risks posed by ingested nanomaterials [15]. In a recent experiment on broiler chickens [16], the addition of Cr at levels of 3 and 6 mg/kg of diet (irrespective of the form used—Cr-Pic or Cr-NPs) reduced abdominal fat and stimulated the blood antioxidant defense system but disturbed liver function and caused histopathological changes in the pancreas and liver. In chickens fed diets with Cr-NPs, a significant degree of hyperemia of the hepatic and pancreatic tissue, as well as extensive foci of fatty degeneration in these organs, were noted [16]. Stępniowska et al. [17] found that the addition of Cr at 3 mg/kg to the diet of broilers, irrespective of the form used—Cr-Pic or Cr-NPs—affected the hormone levels of carbohydrate metabolism (increasing insulin levels and reducing glucagon levels) and adversely affected the antioxidant status of the liver and breast muscle of birds.

The present study postulated that metabolic disturbances in the liver tissue associated with chronic intake of an obesogenic diet could be subsequently alleviated through dietary supplementation with various forms of chromium and/or switching to a low-fat diet. We hypothesized that switching from high-fat dietary habits combined with a pharmacologically relevant dose of chromium supplementation (commonly used picolinate or novel form as nanoparticles; 0.3 mg/kg body weight) would benefit physiological responses in the hepatic status.

## 2. Results

The feeding period consisted of an initial 9 wk and an experimental 9 wk period. During the initial period, C rats were fed a standard low-fat diet (diet C), while the remaining groups (M, F, MP, FP, MN, FN) were subjected to an obesogenic high-fat diet (diet F). The dietary treatments used in the experimental period: Group C, control fed a C-diet; M, fed a C-diet; F, fed an F-diet; MP, fed a C-diet with Cr-Pic supplementation; FP, fed an F-diet with Cr-Pic; MN, fed a C-diet with Cr-NP; FN, fed a F-diet with Cr-NP. Both forms of Cr were added in a dose of 0.3 mg/kg body weight (BW). Two-way ANOVA showed that rats F excelled on the M ones with respect to the final BW, and they had a lower daily feed intake and a lean tissue mass (Table 1). A significant Cr×D interaction showed that the highest daily BW gain was attributed to the F and FP rats, while the lowest was attributed to the M, MP, and MN rats (*p* < 0.05). It should be stressed that the daily BW gain of group FN was significantly higher than in the MN rats but lower versus rats FP (*p* < 0.05). A Cr×D interaction was also noted for fat tissue mass, fat and lean tissue percentage (relative mass), and relative eWAT mass. The fat tissue mass was the highest in the F and FP rats (*p* < 0.05 vs. other experimental groups, including FN rats). When comparing the dietary counterparts fed the same Cr type (i.e., without Cr, with Cr picolinate, and Cr nanoparticles), only the MN and FN groups did not differ significantly between each other regarding the fat tissue mass and fat tissue percentage. The lean tissue percentage for the Cr×D interaction was higher in the FN group than in the FP and F groups (*p* < 0.05). Additionally, the FN group did not differ significantly compared with the MN and MP groups (*p* > 0.05). The Cr×D interaction showed the highest relative mass of the eWAT (epididymal white adipose tissue) in the F rats (*p* < 0.05 vs. all other groups except the FP). The MN and FN counterparts did not differ significantly.

*t*-Test showed that the final BWs of the F, FP, MN, and FN groups were significantly higher than those of the C group (*p* < 0.05; *t*-test) but not in the case for the M and MP groups (*p* > 0.05 vs. C; *t*-test). The daily intake in all experimental groups was significantly lower than that in the C control group (*p* < 0.05 vs. C; *t*-test). Interestingly, the F rats but not the FP and FN ones had lower lean tissue mass when compared to the control C animals (*p* < 0.05; *t*-test).

Two-way ANOVA revealed that with the additional Cr treatment, the dietary Cr nanoparticles enhanced the GSH/GSSG ratio in the liver (*p* < 0.05 vs. treatment W without Cr and with Cr picolinate), while the P (picolinate) treatment decreased the ALT activity in the blood plasma (*p* < 0.05 vs. treatment W; Table 2). The liver fat concentration in the MN group was lower than in the MP group (*p* < 0.05; see significant interaction Cr×D). The Cr×D interaction showed the highest hepatic TC concentration in the M and F groups (*p* < 0.05 vs. MP, FP, MN, FN) and the lowest in the FP, MN, and FN livers (*p* < 0.05 vs. M, F, MP). Interestingly, the hepatic TG concentration in the FP animals was significantly lower than in the FN ones. The addition of Cr-NP to an F-diet did not lower the AST activity in the plasma while the dietary treatment with Cr-Pic caused such decrease.

As indicated by *t*-test, the relative liver mass was higher in the F, MP, and FN groups than in the control C group (*p* < 0.05). Regarding the hepatic fat percentage and plasma ALT activity, the C rats had lower (*p* < 0.05) values than all other groups except the MN group. The hepatic TC and TG concentrations and plasma ALT activity were higher in all experimental groups than in the control C group (*p* < 0.05; *t*-test). The hepatic GHS/GSSG ratio was not decreased significantly in both groups fed diets with Cr-NP addition (MN, FN) as compared to the C rats. 

Two-way ANOVA showed that, irrespective of Cr addition, the feeding regimen with the F obesogenic diet decreased hepatic expression of PPARα and elevated COX-2, HIF-1α, and LOX-1 expression (*p* < 0.05 vs. treatment M; Table 3). Irrespective of the diet type, the dietary addition of both Cr forms increased PPARα expression and decreased COX-2, HIF-1α, and LOX-1 expression in liver tissue. The highest IL-6 level was attributed to the F rats (*p* < 0.05 vs. other groups), while the lowest IL-6 level was attributed to the FN animals (*p* < 0.05 vs. F and FP groups; Figure 1). The *t*-test analyses showed that the C rats had significantly higher PPARα expression in the liver tissue than the M, F, FP, and FN rats. The hepatic expression of COX-2 and LOX-1 was significantly elevated in the experimental groups (except the MN group) compared with that in the C animals. As compared to the C rats, hepatic HIF-1α expression was not increased in the MP, FP, and MN groups, while in the M, F, and FN groups such increase was noted. The F and FP treatment increased liver IL-6 content versus the C rats. Regarding the hepatic IL-10 content, the level in the C group was higher than that in the M, MP, MN, and FN groups (*t*-test; *p* < 0.05).

Dietary treatment with an obesogenic diet caused a significant increase in TC and non-HDL cholesterol concentrations in the plasma irrespective of Cr addition (*p* < 0.05 vs. M; Table 4). The plasma TG concentration and AIP value for the Cr×D interaction was significantly lower in the FP and FN groups than in the F group (*p* < 0.05). 

The *t*-test analyses showed that the C group had lower plasma TC and nHDL concentrations than all experimental groups (*p* < 0.05). As compared to rats C, the plasma HDL concentration was significantly decreased in all experimental groups, but not in the FN one. The AIP values were not enhanced by treatments FP and FN versus the control C animals (*t*-test). As compared to the C group, the glucose concentration in the blood plasma was higher only in F and FN rats (*p* < 0.05).

## 3. Discussion

Diets rich in fats, mostly saturated fatty acids, and low in dietary fiber increase the risk of obesity and thus disorders mostly associated with a spectrum of liver abnormalities, known as nonalcoholic fatty liver disease (NAFLD). These liver disorders are often characterized by an increased intrahepatic triglyceride content with inflammation and a higher risk of dyslipidemia (high plasma TG and/or low plasma HDL-cholesterol concentrations) [18]. In the present study, rats fed a high-fat/low-fiber (obesogenic) diet showed similar disorders, e.g., elevation of body mass gain, fat mass, intrahepatic fat, triglycerides, cholesterol content, and hepatic inflammatory parameters. An efficient way to regulate diet-induced obesity-related disorders is to switch back to a balanced diet. Braga et al. (2020) [19], in a nutritional study on mice, demonstrated that the development of metabolic impairments (increase in body mass gain and hyperglycemia) following a high-fat diet might be improved by switching back to a balanced diet. Additionally, in our study, the strongest effect against increased fat mass, body mass gain, and final body mass was observed when the obesogenic diet was changed to a standard diet. However, this dietary change was insufficient to regulate obesity-related liver disorders. To support the health-promoting effect of the standard diet, two types of chromium were added: nanoparticles (nonoxide form) and picolinate. To our best knowledge, this nutritional study is the first on rats that compares the impact of supplementation with these two chromium forms on the development of fatty liver disorders. The previous own experiments showed negative oxidation consequences of dietary Cr-NPs (0.3 mg kg BW) in different organs of rat, e.g., in the small intestine, liver, brain, heart, and thoracic aorta [20,21,22]. That dietary pharmacologically relevant dose of chromium nanoparticles also caused some disturbances in blood distribution of minerals [23], however without excessive accumulation of chromium in the rats’ tissues, including the liver [24]. Among the examined forms of chromium, only a standard diet supplemented (0.3 mg/kg of BW) with nanoparticles considerably mitigated liver disorders induced by an obesogenic diet. In this group, a reduction in total cholesterol, triglycerides, and fat content and an increase in liver antioxidant potential were observed by elevating the GSH/GSSG ratio. However, regardless of the chromium type, the plasma lipid profile was not improved. Fatima and Ahmad (2019) [25], in a toxicological study on Wistar rats, showed that administration of chromium oxide nanoparticles at doses of 1–2 mg/kg BW exerted hepatotoxic effects. Furthermore, in vitro studies showed that exposure to chromium oxide nanoparticles enhanced the production of intracellular reactive oxygen species [26,27]. Compared with our nutritional study, these unfavorable effects might be related to different chemical structures and the high dose of chromium oxide nanoparticles. Opposite effects were observed in nutritional studies using chromium picolinate. Chromium picolinate supplementation was favorably associated with the regulation of lipid metabolism, blood lipid profiles, and inflammatory biomarkers in organisms with obesity-related disorders [8,9,28]. In our study, a stronger effect against disorders induced by an obesogenic diet was observed when the diet was supplemented with chromium nanoparticles.

The possible molecular mechanisms involved in the observed changes in the livers of rats fed a standard diet supplemented with chromium nanoparticles are associated with hepatic downregulation of HIF-1α and upregulation of PPARα. These molecular factors are key players in the development of NAFLD [29]. HIF-1α activation alters mitochondrial respiratory function and metabolism, affecting energetic and redox homeostasis, while PPARα activation is associated with the regulation of fatty acid metabolism, oxidative catabolism of fatty acids, and inflammatory mechanisms [30,31]. The lowest expression levels of COX-2 and LOX-1, which are responsible for activating inflammation and oxidative stress mechanisms in the liver, were found in the livers of rats fed a diet with chromium nanoparticles [30,32]. Additionally, functional expression of LOX-1 in hepatocytes may activate primary neutral sterol transporters in hepatobiliary processes, increasing hepatocyte lipid uptake [33]. Animals from this group also had the lowest values of aminotransferase activity in the serum, a factor indicating the development of inflammation in the liver. Other studies have shown that COX-2 upregulation following experimental nutritional steatohepatitis is suppressed by PPARα agonists, and this effect has been ascribed to the ability of activated PPARα to interfere with the NF-κB signaling pathway [34]. This mechanism might partially explain the molecular mechanisms by which chromium nanoparticles regulate obesity-related liver disorders.

In the present study, the effect of supplementation with chromium without switching from an obesogenic to a balanced diet was also examined. In this group, the regulatory effect against obesity-related disorders was not as effective as that in rats fed a standard diet. This part of the experiment showed that adding chromium nanoparticles at a pharmacologically relevant dose of 0.3 mg/kg BW to the obesogenic diet might regulate BW gain, fat mass, and epididymal white adipose tissue. These changes were not observed in rats fed with the standard low-fat diet. The reason might be that the favorable effects of chromium nanoparticles were masked by the strong impact of switching from an obesogenic to a standard diet. The combination with an obesogenic diet showed that chromium, a key microelement involved in the metabolism of fats in nanoparticle form, might have higher accessibility and thus more effectively interact with mechanisms against some obesity-related disorders [3]. The obesogenic diet increased the accumulation of liver lipids and activated molecular mechanisms related to the upregulation of COX-2 and thus the development of proinflammatory processes and oxidative stress [30,34]. The activation of hepatic HIF-1α inhibits the mechanism associated with the activation of PPARα, possibly explaining the observed increased levels of fat and cholesterol in the livers of rats fed an obesogenic diet. Fatty liver-induced mechanisms are linked to the activation of LOX-1, which is partly responsible for the transport of cholesterol from the liver [33]. These hepatic mechanisms leading to the development of fatty liver disorders were regulated to some extent by supplementation with both forms of chromium. These microelements, regardless of type, considerably reduced fat, cholesterol, and the triglyceride content in the liver and improved the plasma lipid profile by decreasing triglycerides and thus the atherogenic index of plasma. Zhou et al. (2013) [35] in a nutritional study on sheep showed that chromium supplementation can improve lipid profiles by decreasing triglyceride synthesis and enhancing adipose tissue decomposition. In addition to the obesogenic diet, chromium nanoparticles increased liver antioxidative potential by increasing the GSH/GSSG ratio. Additionally, both chromium forms considerably lowered the expression levels of COX-2, HIF-1α, and LOX-1 and upregulated PPARα in rats fed an obesogenic diet. These changes in hepatic molecular mechanisms might explain the observed decreasing level of IL-6 and lower activity of ALT in the plasma. These inflammatory factors play a fundamental role in the onset and progression of NAFLD, promoting oxidative stress, hepatic inflammation, cell necrosis, apoptosis, and liver fibrosis [36].

## 4. Materials and Methods

### 4.1. Chromium Form Characterization

Chromium nanoparticles (Cr-NPs) were purchased from Sky Spring Nanomaterials Inc. (Houston, TX, USA), with a purity of 999 g/kg, 60–80 nm in size (nanopowder), spherical morphology, a specific surface area of 6–8 m^2^/g, 0.15 g/cm^3^ bulk density, and 8.9 g/cm^3^ true density. Chromium picolinate (Cr-Pic), with a purity >980 g/kg, was purchased from Sigma-Aldrich Co. (Poznań, Poland).

### 4.2. The In Vivo Study

The in vivo experiment was performed using 84 outbred male Wistar rats (Cmdb:Wi CMDB) fed a standard low-fat or high-fat/low-fiber (obesogenic) semi-purified rat diet without or with the dietary addition of two chromium (nanoparticles and picolinate; Cr-NP and Cr-Pic, respectively) forms (Table 5). The study schema consisted of two periods, initial and experimental, lasting 9 weeks each (Table 6). During the initial 9 wk period, the rats aged 6 weeks were randomly assigned to the control group (*n* = 12) fed a standard low-fat C diet and the HF group (*n* = 72) fed an obesogenic diet. After the initial period, the rats from the control C group were fed the same standard low-fat C diet for the subsequent 9 weeks of the experimental period. The HF rats were then randomly divided into 6 groups with *n* = 12 per group. The M group was subjected to a standard low-fat diet (the same as in the control group; this treatment imitates a change in the eating habits of an obese consumer without chromium supplementation—i.e., switching from a high energy density diet to a low-fat diet); by contrast, the F group was further fed the obesogenic diet for a subsequent 9 weeks (illustrating no changes in the eating habits of an obese consumer); the MP group was fed a standard low-fat diet with supplementation of Cr-Pic (imitating a switch from an obesogenic diet to a low-fat diet along with common Cr form support); the MN group was fed a standard low-fat diet with supplementation of Cr-NP (mirroring a switching from an obesogenic diet to a low-fat diet along with novel Cr form support); the FP group was subjected to an obesogenic diet with Cr-Pic supplementation (no changes were made in the main dietary patterns, but an obese consumer is aware of unhealthy eating habits and will take common Cr supplements); the FN group in turn was fed obesogenic diet with Cr-NP supplementation (no changes were made in the main dietary patterns, but an obese consumer is aware of unhealthy eating habits and will take novel Cr supplements). The C, MP, and MN groups were fed diets with 15.1, 21.5, and 63.4 kcal% originating from protein, fat, and carbohydrates, respectively. In the case of diets F, FP, and FN those values were 11.6, 48.8, and 39.6 kcal% for protein, fat, carbohydrates, respectively. The amount of chromium administered to each rat (MP, FP, MN, and FN groups) was 0.3 mg/kg body weight (BW), selected according to the EFSA NDA Panel (2014) [37]. The applied dosage should be considered the pharmacologically relevant dose of additional Cr in the diet. Regarding the safety of the operator preparing the experimental diets with the Cr-NP preparation, both Cr sources (to maintain comparable conditions) were added to the diet as an emulsion together with dietary rapeseed oil rather than in the mineral mixture.

The FN group in turn illustrates no changes in the main dietary patterns but an obese consumer who is aware of unhealthy eating habits and starts taking novel Cr supplements. All animal care and experimental protocols complied with the current laws governing animal experimentation in the Republic of Poland and by an ethical committee according to the European Convention for the Protection of Vertebrate Animals used for Experimental and other Scientific Purposes, Directive 2010/63/EU for animal experiments, and they were approved by the National Ethics Committee for Animal Experiments (Approval No. 73/2021). Rats were housed randomly and individually in stainless steel cages under a stable temperature (22 ± 1 °C), relative humidity 60 ± 5%, a 12 h light–dark cycle, and a ventilation rate of 15 air changes per hour. All animals throughout the study were monitored daily for any abnormal rat behavior and any indicators of animal fear, distress, pain, or anxiety to respect the humane endpoints in animal research. For 18 weeks (9 wk initial and 9 wk experimental period), the rats had free access to tap water and semi-purified diets, which were prepared and then stored at 4 °C in hermetic containers until the end of the experiment. The diets were modifications of a casein diet for laboratory rodents recommended by the American Institute of Nutrition [38].

All physiological measurements were performed for each animal separately (*n* = 12 for each group). Experimental groups were monitored for body weight gain and feed intake. One day before study termination, the living rats were subjected to time-domain nuclear magnetic resonance using a minispec LF 90II analyser (Bruker, Karlsruhe, Germany) to determine the fat and lean tissue mass. The minispec transmits various radio frequency pulse sequences into soft tissues to reorient the nuclear magnetic spins of the hydrogen and then detects radio frequency signals generated by the hydrogen spins from these tissues. The contrast in relaxation times of the hydrogen spins found between adipose tissue and water-rich tissues is used to estimate fat and lean body mass. At the end of the experiment, the rats were fasted for 12 h and anesthetized i.p. with ketamine and xylazine (K, 100 mg/kg BW; X, 10 mg/kg BW) according to the recommendations for anesthesia and euthanasia of experimental animals. Next, after laparotomy, blood samples were taken from the caudal vena cava into heparinized tubes, and then the rats were euthanized by cervical dislocation. After that, the liver and epididymal white adipose tissue (eWAT) were dissected and weighed. Blood plasma was obtained by centrifugation (350×g, 10 min, 4 °C) and kept frozen at −70 °C until analysis. The liver samples were immediately frozen in liquid nitrogen and then kept under similar conditions as plasma.

### 4.3. Laboratory Analyses

After storage of the liver at −80 °C, IL-6 and IL-10 expression was determined using commercial ELISA kits (Sigma-Aldrich, St. Louis, MO, USA). Hepatic reduced glutathione (GSH) and oxidized glutathione (GSSG) concentrations were determined using an enzymatic recycling method described by Rahman et al. (2006) [39]. Liver lipids were extracted according to Folch et al. (1957) [40]. Following extraction, total cholesterol (TC) and triglyceride (TG) concentrations were determined enzymatically using commercial kits (Alpha Diagnostics Ltd., Warsaw, Poland). Total RNA was extracted from liver samples using TRI Reagent (Sigma-Aldrich). The quantity and quality of RNA were checked using a NanoDrop 1000 instrument (Thermo Scientific, Waltham, MA, USA). cDNA was synthesized from 500 ng of total RNA using a High-Capacity cDNA Reverse Transcription Kit with RNase Inhibitor (Applied Biosystem, Waltham, MA, USA). To measure the hepatic levels of peroxisome proliferator-activated receptor alpha (PPARα), cyclooxygenase 2 (COX-2), hypoxia-inducible Factor 1 alpha (HIF-1α) and lectin-like oxidized low-density lipoprotein (LDL) receptor-1 (LOX-1) mRNA expression, single tube TaqMan^®^ Gene Expression Assays (Life Technologies, Carlsbad, CA, USA) were used. Amplification was performed using a 7900HT Fast Real-Time PCR System under the following conditions: initial denaturation for 10 min at 95 °C, followed by 40 cycles of 15 s at 95 °C and 1 min at 60 °C. Each run included a standard curve based on aliquots of pooled liver RNA. All samples were analyzed in duplicate. The mRNA expression levels were normalized to the β-actin (ACTB) reference gene. In the blood plasma, the triglycerides (TG) and total cholesterol (TC), fraction of HDL cholesterol (HDL), glucose concentrations and activity of AST, ALP and ALT were estimated using a biochemical analyzer (Pentra C200; Horiba, Tokyo, Japan). The atherogenic index of plasma (AIP) was calculated using the formula log(TG/HDL).

### 4.4. Statistics

STATISTICA software, version 12.0 (StatSoft Corp., Krakow, Poland), was applied to determine whether variables differed among treatment groups. Two-way ANOVA and Student’s *t*-test were used to analyze the results. Two-way ANOVA was applied to assess the effects of the main factors: diet type (D; low-fat, high-fat/low fiber), additional Cr type (Cr; without, picolinate, nanoparticles), and the interaction between them (Cr×D). When ANOVA indicated significant treatment effects, the means were evaluated using Duncan’s multiple range test. The data were checked for normality before statistical analysis was performed. Differences with *p* ≤ 0.05 were considered significant.

## 5. Conclusions

The disorders induced by the obesogenic diet were effectively mitigated when the diet was switched to the standard (low fat) diet. Supplementation of the standard diet with chromium nanoparticles considerably enhanced favorable effects against the development of fatty liver disorders. This combination exerted the strongest reduction in fat content and cholesterol in the liver. In this group, a favorable antioxidative effect was also observed through elevation of GSH/GSSG in the liver as well as reduction of ALT activity in plasma and level of IL-6 in the liver. The mechanisms involved in the regulation of lipid metabolism, oxidative stress, and development of inflammation might be associated with lower expression of COX-2, HIF-1α, and LOX-1 and upregulation of PPARα. In the group where the obesogenic diet was not switched to the standard diet, supplementation with chromium nanoparticles exerted similar favorable effects against obesity-related disorders but was not as efficient as that observed in the group with the standard diet. Thus, the findings of the present study indicate that switching from obesogenic dietary habits together with chromium nanoparticle supplementation benefit physiological responses in the hepatic status, increasing regulatory effects against obesity-related disorders. Furthermore, this experiment was performed in model animals; therefore, the use of chromium nanoparticles as a dietary supplement or functional additive to food should also be verified in human studies.

## Figures and Tables

**Figure 1 ijms-24-02940-f001:**
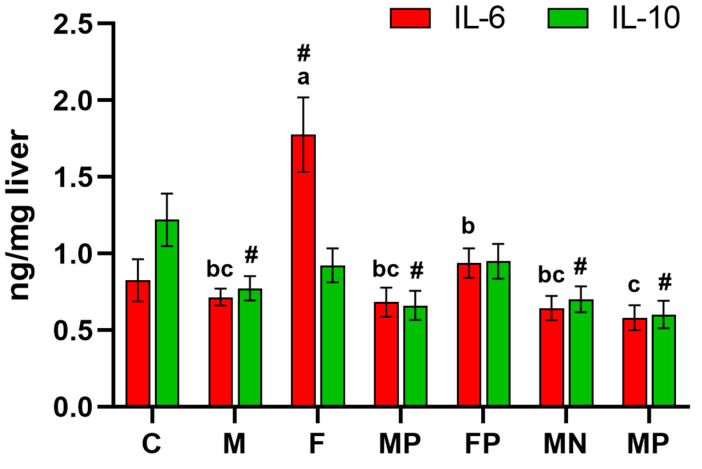
Inflammatory cytokines in the liver of rats fed the experimental diets (*n* = 12 per group). ^a–c^ Mean values with unlike superscript letters are shown to be significantly different (*p* < 0.05) in regard to the experimental groups M, F, MP, FP, MP, MN. Additionally, each experimental group was compared with the control C group using *t*-test (# indicates a significant difference versus the C group). IL-6, interleukin 6; IL-10, interleukin 10.

**Table 1 ijms-24-02940-t001:** Body weight, feed intake and NMR (nuclear magnetic resonance) parameters in rats fed experimental diets (*n* = 12 per group) *.

	Initial BW	Final BW	BW Gain	Intake	Fat Mass	Lean Mass	Fat	Lean	eWAT
	G	g	g/Day	g/Day	g	G	%	%	g/100 g BW
Control C	409	469	1.04	18.5	74.1	118	31.6	50.2	3.65
2-way ANOVA:									
M	461 ^#^	475	0.245 ^c#^	16.7 ^#^	77.0 ^c^	118	32.5 ^d^	49.6 ^a^	3.78 ^d^
F	462 ^#^	542 ^#^	1.341 ^a^	15.5 ^#^	118 ^a#^	109 ^#^	43.3 ^a#^	40.4 ^c#^	5.15 ^a#^
MP	462 ^#^	481	0.308 ^c#^	17.1 ^#^	79.9 ^c^	118	33.2 ^d^	48.9 ^ab^	3.90 ^d^
FP	461 ^#^	535 ^#^	1.262 ^a^	15.5 ^#^	108 ^a#^	113	40.2 ^b#^	42.3 ^c#^	4.73 ^ab#^
MN	462 ^#^	484 ^#^	0.356 ^c#^	16.7 ^#^	82.0 ^bc^	117	33.8 ^cd^	48.7 ^ab^	3.97 ^cd^
FN	453 ^#^	510 ^#^	0.977 ^b^	14.4 ^#^	94.1 ^b#^	116	36.7 ^c#^	45.8 ^b#^	4.40 ^bc#^
SEM	4.139	5.508	0.058	0.183	2.302	1.21	0.604	0.540	0.080
Cr (additional)									
W (without)	461	509	0.793	16.1	97.3	113	37.8	45.0	4.46
P (picolinate)	461	508	0.784	16.3	94.0	115	36.7	45.6	4.31
N (nano)	457	497	0.666	15.5	88.0	117	35.2	47.2	4.18
*p*-value	0.909	0.601	0.292	0.078	0.101	0.478	0.059	0.102	0.223
D (Diet type)									
M	461	479 ^b^	0.303	16.8 ^a^	79.6	117 ^a^	33.1	49.0	3.88
F	459	529 ^a^	1.19	15.1 ^b^	106	112 ^b^	40.0	42.8	4.76
*p*-value	0.767	<0.001	<0.001	<0.001	<0.001	0.047	<0.001	<0.001	<0.001
Interaction Cr×D									
*p*-value	0.864	0.294	0.029	0.332	0.006	0.428	0.002	0.016	0.015

* The feeding period consisted of an initial 9 wk and an experimental 9 wk period. During the initial period, group C rats were fed a diet with 8% rapeseed oil and 64% maize starch (low-fat diet C), while the remaining groups were subjected to an obesogenic diet with 8% rapeseed oil, 17% lard, and 52% maize starch (high-fat diet F). The dietary treatments used in the experiment were as follows: Group C, control, fed a C-diet during the initial and the experimental periods; M, fed an F-diet during the initial period and a C-diet during the experimental period; F, fed an F-diet during the initial and the experimental periods; MP, fed an F-diet during the initial period and a C-diet with supplementation of chromium picolinate (Cr-Pic; 0.3 mg/kg BW) during the experimental period; FP, fed an F-diet during the initial period and an F-diet with supplementation of Cr-Pic (0.3 mg/kg BW) during the experimental period; MN, fed an F-diet during the initial period and a C-diet with supplementation of chromium nanoparticles (Cr-NP; 0.3 mg/kg BW) during the experimental period; FN, fed an F-diet during the initial period and an F-diet with supplementation of Cr-NP (0.3 mg/kg BW) during the experimental period. W, treatment (*n* = 24) without Cr supplementation; P, treatment (*n* = 24) with Cr-Pic supplementation; *n*, treatment (*n* = 24) with Cr-NP supplementation; M, treatment (*n* = 36) with a standard C-diet; F, treatment (*n* = 36) with an obesogenic F-diet. ^a–d^ Mean values within a column with unlike superscript letters are shown to be significantly different (*p* < 0.05); differences among the groups (M, F, MP, FP, MP, MN) are indicated with superscripts only in the case of a statistically significant interaction Cr×D (*p* < 0.05). Additionally, each experimental group was compared with the control C group using *t*-test (# indicates a significant difference versus the C group). eWAT, epididymal white adipose tissue; BW, body weight; SEM, pooled standard error of mean (standard deviation for all rats divided by the square root of the rat number, *n* = 84).

**Table 2 ijms-24-02940-t002:** Hepatic functional parameters and blood plasma AST, ALT and ALP activity in rats fed experimental diets (*n* = 12 per group) *.

	Weight	Fat	TC	TG	GSH/GSSG	AST	ALT	ALP
	g/100 g BW	%	mg/g	mg/g		U/L	U/L	U/L
Control C	3.07	16.8	12.3	15.5	3.57	84.9	36.6	80.2
2-way ANOVA:								
M	3.29	24.6 ^c#^	25.5 ^a#^	32.5 ^b#^	2.70 ^#^	105 ^b#^	80.1 ^#^	90.9 ^#^
F	3.56 ^#^	36.9 ^a#^	24.2 ^a#^	42.9 ^a#^	2.39 ^#^	132 ^a#^	96.2 ^#^	91.1^#^
MP	3.39 ^#^	24.4 ^c#^	22.1 ^b#^	30.9 ^bc#^	2.90 ^#^	106 ^b#^	55.2 ^#^	85.3
FP	3.30	29.8 ^b#^	16.2 ^c#^	21.1 ^d#^	2.80 ^#^	106 ^b#^	69.6 ^#^	77.9
MN	3.13	19.1 ^d^	15.7 ^c#^	22.6 ^d#^	3.43	93.9 ^b^	60.7 ^#^	90.4
FN	3.53 ^#^	31.0 ^b#^	15.3 ^c#^	27.6 ^c#^	3.05	139 ^a#^	89.6 ^#^	83.5
SEM	0.044	3.490	2.508	3.686	0.399	3.184	3.017	1.587
Cr (additional)								
W (without)	3.42	30.7	24.8	37.6	2.54 ^b^	118	88.1 ^a^	90.9
P (picolinate)	3.34	27.1	19.1	25.9	2.85 ^b^	106	62.4 ^b^	81.5
N (nano)	3.33	25.1	15.5	25.1	3.24 ^a^	116	75.1 ^ab^	86.9
*p*-value	0.672	<0.001	<0.001	<0.001	0.001	0.181	<0.005	0.084
D (Diet type)								
M	3.26 ^b^	22.7	21.1	28.7	3.01	101	65.3 ^b^	88.8
F	3.46 ^a^	32.6	18.6	30.5	2.75	125	85.1 ^a^	84.1
*p*-value	0.037	<0.001	<0.001	0.158	0.083	<0.001	<0.001	0.171
Interaction Cr×D								
*p*-value	0.086	0.029	0.001	<0.001	0.717	0.010	0.442	0.598

* The feeding period consisted of an initial 9 wk and an experimental 9 wk period. During the initial period, group C rats were fed a diet with 8% rapeseed oil and 64% maize starch (low-fat diet C), while the remaining groups were subjected to an obesogenic diet with 8% rapeseed oil, 17% lard, and 52% maize starch (high-fat diet F). The dietary treatments used in the experiment were as follows: Group C, control, fed a C-diet during the initial and the experimental periods; M, fed an F-diet during the initial period and a C-diet during the experimental period; F, fed an F-diet during the initial and the experimental periods; MP, fed an F-diet during the initial period and a C-diet with supplementation of chromium picolinate (Cr-Pic; 0.3 mg/kg BW) during the experimental period; FP, fed an F-diet during the initial period and an F-diet with supplementation of Cr-Pic (0.3 mg/kg BW) during the experimental period; MN, fed an F-diet during the initial period and a C-diet with supplementation of chromium nanoparticles (Cr-NP; 0.3 mg/kg BW) during the experimental period; FN, fed an F-diet during the initial period and an F-diet with supplementation of Cr-NP (0.3 mg/kg BW) during the experimental period. W, treatment (*n* = 24) without Cr supplementation; P, treatment (*n* = 24) with Cr-Pic supplementation; *n*, treatment (*n* = 24) with Cr-NP supplementation; M, treatment (*n* = 36) with a standard C-diet; F, treatment (*n* = 36) with an obesogenic F-diet. ^a–d^ Mean values within a column with unlike superscript letters are shown to be significantly different (*p* < 0.05); differences among the groups (M, F, MP, FP, MP, MN) are indicated with superscripts only in the case of a statistically significant interaction Cr×D (*p* < 0.05). Additionally, each experimental group was compared with the control C group using *t*-test (# indicates a significant difference versus the C group). AST, aspartate transaminase; ALT, alanine transaminase; ALP, alkaline phosphatase; TC, total cholesterol; TG, triglycerides; GSH, reduced glutathione; GSSG, oxidized glutathione; SEM, pooled standard error of mean (standard deviation for all rats divided by the square root of rat number, *n* = 84).

**Table 3 ijms-24-02940-t003:** Hepatic mRNA expression (relative expression normalized to β-actin × 10) of selected factors in the liver of rats fed experimental diets (*n* = 12 per group) *.

	PPARα	COX-2	HIF-1α	LOX-1
Control C	21.6	7.51	8.05	6.54
2-way ANOVA:				
M	14.7 ^#^	11.1 ^#^	9.59 ^#^	12.4 ^#^
F	7.57 ^#^	13.0 ^#^	11.0 ^#^	22.9 ^#^
MP	16.8	10.4 ^#^	8.49	9.88 ^#^
FP	14.4 ^#^	10.4 ^#^	8.63	14.5 ^#^
MN	21.7	8.36	8.47	6.23
FN	12.7 ^#^	11.6 ^#^	10.1 ^#^	14.0 ^#^
SEM	2.094	1.385	1.228	1.651
Cr (additional)				
W (without)	11.1 ^b^	12.0 ^a^	10.3 ^a^	17.6 ^a^
P (picolinate)	15.6 ^a^	10.4 ^b^	8.56 ^b^	12.2 ^b^
N (nano)	17.2 ^a^	10.0 ^b^	9.27 ^b^	10.1 ^b^
*p*-value	0.007	0.047	0.001	<0.001
D (Diet type)				
M	17.8 ^a^	9.96 ^b^	8.85 ^b^	9.51 ^b^
F	11.6 ^b^	11.7 ^a^	9.90 ^a^	17.1 ^a^
*p*-value	<0.001	0.013	0.005	<0.001
Interaction Cr×D				
*p*-value	0.220	0.146	0.200	0.186

* The feeding period consisted of an initial 9 wk and an experimental 9 wk period. During the initial period, group C rats were fed a diet with 8% rapeseed oil and 64% maize starch (low-fat diet C), while the remaining groups were subjected to an obesogenic diet with 8% rapeseed oil, 17% lard, and 52% maize starch (high-fat diet F). The dietary treatments used in the experiment were as follows: Group C, control, fed a C-diet during the initial and the experimental periods; M, fed an F-diet during the initial period and a C-diet during the experimental period; F, fed an F-diet during the initial and the experimental periods; MP, fed an F-diet during the initial period and a C-diet with supplementation of chromium picolinate (Cr-Pic; 0.3 mg/kg BW) during the experimental period; FP, fed an F-diet during the initial period and an F-diet with supplementation of Cr-Pic (0.3 mg/kg BW) during the experimental period; MN, fed an F-diet during the initial period and a C-diet with supplementation of chromium nanoparticles (Cr-NP; 0.3 mg/kg BW) during the experimental period; FN, fed an F-diet during the initial period and an F-diet with supplementation of Cr-NP (0.3 mg/kg BW) during the experimental period. W, treatment (*n* = 24) without Cr supplementation; P, treatment (*n* = 24) with Cr-Pic supplementation; *n*, treatment (*n* = 24) with Cr-NP supplementation; M, treatment (*n* = 36) with a standard C-diet; F, treatment (*n* = 36) with an obesogenic F-diet. ^a,b^ Mean values within a column with unlike superscript letters are shown to be significantly different (*p* < 0.05); differences among the groups (M, F, MP, FP, MP, MN) are indicated with superscripts only in the case of a statistically significant interaction Cr×D (*p* < 0.05). Additionally, each experimental group was compared with the control C group using *t*-test (# indicates a significant difference versus the C group). PPARα, peroxisome proliferator-activated receptor alpha; COX-2, cyclooxygenase 2; HIF-1α, hypoxia-inducible Factor 1 alpha; LOX-1, lectin-like oxidized low-density lipoprotein (LDL) receptor-1; SEM, pooled standard error of the mean (standard deviation for all rats divided by the square root of the rat number, *n* = 84).

**Table 4 ijms-24-02940-t004:** Blood plasma parameters in rats fed experimental diets (*n* = 12 per group) *.

	TC	HDL	nHDL	TG	AIP	GL
	mmol/L	mmol/L	mmol/L	mmol/L	log(TG/HDL)	mmol/L
Control C	1.96	0.488	1.47	1.54	0.497	13.2
2-way ANOVA:						
M	2.16 ^#^	0.414 ^#^	1.75 ^#^	1.49 ^ab^	0.547 ^ab^	13.7
F	2.46 ^#^	0.407 ^#^	2.06 ^#^	1.74 ^a^	0.633 ^a#^	15.2 ^#^
MP	2.18 ^#^	0.410 ^#^	1.77 ^#^	1.56 ^ab^	0.577 ^ab#^	13.3
FP	2.26 ^#^	0.408 ^#^	1.85 ^#^	1.37 ^b^	0.513 ^b^	13.4
MN	2.16 ^#^	0.415 ^#^	1.74 ^#^	1.60 ^ab^	0.580 ^ab#^	13.8
FN	2.27 ^#^	0.429	1.84 ^#^	1.33 ^b^	0.489 ^b^	15.3 ^#^
SEM	0.032	0.008	0.030	0.038	0.012	0.164
Cr (additional)						
W (without)	2.31	0.410	1.90	1.61	0.590	14.4 ^a^
P (picolinate)	2.22	0.408	1.80	1.46	0.545	13.3 ^b^
N (nano)	2.21	0.422	1.79	1.46	0.534	14.0 ^ab^
*p*-value	0.351	0.773	0.217	0.193	0.151	0.031
D (Diet type)						
M	2.16 ^b^	0.413	1.75 ^b^	1.54	0.568	13.6
F	2.33 ^a^	0.414	1.91 ^a^	1.47	0.545	14.2
*p*-value	0.012	0.933	0.004	0.383	0.352	0.062
Interaction Cr×D						
*p*-value	0.301	0.855	0.176	0.021	0.008	0.209

* The feeding period consisted of an initial 9 wk and an experimental 9 wk period. During the initial period, group C rats were fed a diet with 8% rapeseed oil and 64% maize starch (low-fat diet C), while the remaining groups were subjected to an obesogenic diet with 8% rapeseed oil, 17% lard, and 52% maize starch (high-fat diet F). The dietary treatments used in the experiment were as follows: Group C, control, fed a C-diet during the initial and the experimental periods; M, fed an F-diet during the initial period and a C-diet during the experimental period; F, fed an F-diet during the initial and the experimental periods; MP, fed an F-diet during the initial period and a C-diet with supplementation of chromium picolinate (Cr-Pic; 0.3 mg/kg BW) during the experimental period; FP, fed an F-diet during the initial period and an F-diet with supplementation of Cr-Pic (0.3 mg/kg BW) during the experimental period; MN, fed an F-diet during the initial period and a C-diet with supplementation of chromium nanoparticles (Cr-NP; 0.3 mg/kg BW) during the experimental period; FN, fed an F-diet during the initial period and an F-diet with supplementation of Cr-NP (0.3 mg/kg BW) during the experimental period. W, treatment (*n* = 24) without Cr supplementation; P, treatment (*n* = 24) with Cr-Pic supplementation; *n*, treatment (*n* = 24) with Cr-NP supplementation; M, treatment (*n* = 36) with a standard C-diet; F, treatment (*n* = 36) with an obesogenic F-diet. ^a,b^ Mean values within a column with unlike superscript letters are shown to be significantly different (*p* < 0.05); differences among the groups (M, F, MP, FP, MP, MN) are indicated with superscripts only in the case of a statistically significant interaction Cr×D (*p* < 0.05). Additionally, each experimental group was compared with the control C group using *t*-test (# indicates a significant difference versus the C group). TC, total cholesterol; HDL, high density lipoprotein; nHDL, non HDL cholesterol; TG, triglycerides; AIP, atherogenic index of plasma [log(TG/HDL)]; GL, glucose; SEM, pooled standard error of mean (standard deviation for all rats divided by the square root of rat number, *n* = 84).

**Table 5 ijms-24-02940-t005:** Diets used in the experiment (%).

	C	F	MP	FP	MN	FN
Casein ^1^	14.8	14.8	14.8	14.8	14.8	14.8
DL-methionine	0.2	0.2	0.2	0.2	0.2	0.2
Cellulose ^2^	8	3	8	3	8	3
Choline chloride	0.2	0.2	0.2	0.2	0.2	0.2
Cholesterol	0.3	0.3	0.3	0.3	0.3	0.3
Vitamin mix ^3^	1	1	1	1	1	1
Mineral mix ^4^	3.5	3.5	3.5	3.5	3.5	3.5
Maize starch ^5^	64	52	64	52	64	52
Rapeseed oil	8	8	8 (with Cr-Pic)	8 (with Cr-Pic)	8 (with Cr-NP)	8 (with Cr-NP)
Lard	0	17	0	17	0	17
Calculated nutritional value						
Protein, kcal%	15.1	11.6	15.1	11.6	15.1	11.6
Fat, kcal%	21.5	48.8	21.5	48.8	21.5	48.8
Carbohydrates, kcal%	63.4	39.6	63.4	39.6	63.4	39.6

^1^ Casein preparation: crude protein, 89.7%; crude fat, 0.3%; ash, 2.0; water, 8.0%. ^2^ α-Cellulose (SIGMA, Poznan, Poland), main source of dietary fiber. ^3^ AIN-93G-VM [38], g/kg mix: 3.0 nicotinic acid, 1.6 Ca pantothenate, 0.7 pyridoxine-HCl, 0.6 thiamin-HCl, 0.6 riboflavin, 0.2 folic acid, 0.02 biotin, 2.5 vitamin B-12 (cyanocobalamin, 0.1% in mannitol), 15.0 vitamin E (all-rac-α-tocopheryl acetate, 500 IU/g), 0.8 vitamin A (all-trans-retinyl palmitate, 500,000 IU/g), 0.25 vitamin D-3 (cholecalciferol, 400,000 IU/g), 0.075 vitamin K-1 (phylloquinone), 974.655 powdered sucrose. ^4^ Mineral mix, g/kg mix: 357 calcium carbonate anhydrous CaCO_3_, 196 dipotassium phosphate K_2_HPO_4_, 70.78 potassium citrate C_6_H_5_K_3_O_7_, 74 sodium chloride NaCl, 46.6 potassium sulfate K_2_SO_4_, 24 magnesium oxide MgO, 18 microelement mixture, starch to 1 kg = 213.62. Microelement mixture, g/kg mix: 31 iron (III) citrate (16.7% Fe), 4.5 zinc carbonate ZnCO_3_ (56% Zn), 23.4 manganese (II) carbonate MnCO_3_ (44.4% Mn), copper carbonate CuCO_3_ (55.5% Cu), 0.04 potassium iodide KI, citric acid C_6_H_8_O_7_ to 100 g. ^5^ Maize starch preparation: crude protein, 0.6%; crude fat, 0.9%; ash, 0.2%; total dietary fiber, 0%; water, 8.8%.

**Table 6 ijms-24-02940-t006:** Schema of the experimental groups * (*n* = 12).

	Group Control C	Group M	Group F	Group MP	Group FP	Group MN	Group FN
Initial 9 wk period	Diet C	Diet F	Diet F	Diet F	Diet F	Diet F	Diet F
Experimental 9 wk period	Diet C	Diet C	Diet F	Diet MP	Diet FP	Diet MN	Diet FN
Initial 9 wk period	Without Cr supplementation	Without Cr supplementation	Without Cr supplementation	Without Cr supplementation	Without Cr supplementation	Without Cr supplementation	Without Cr supplementation
Experimental 9 wk period	Without Cr supplementation	Without Cr supplementation	Without Cr supplementation	Cr-Pic (0.3 mg/kg BW)	Cr-Pic (0.3 mg/kg BW)	Cr-NP (0.3 mg/kg BW)	Cr-NP (0.3 mg/kg BW)

* The control C group imitates healthy eating habits in a consumer; the M group imitates a change in the eating habits of an obese consumer without chromium supplementation, i.e., switching from a high energy density diet to a low-fat diet; the F group illustrates no changes in the eating habits of an obese consumer; the MP group imitates a switch from an obesogenic diet to a low-fat diet along with common Cr form support; the FP group reflects no changes in the main dietary patterns but an obese consumer who is aware of unhealthy eating habits and decides to take common Cr supplements; the MN group mirrors a switch from an obesogenic diet to a low-fat diet along with novel Cr form support.

## Data Availability

Data supporting reported results are available on request.

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
