# Peer review of "Chromium Nanoparticles Together with a Switch Away from High-Fat/Low-Fiber Dietary Habits Enhances the Pro-Healthy Regulation of Liver Lipid Metabolism and Inflammation in Obese Rats"

_ijms, 2023, doi:10.3390/ijms24032940_

Round 1

Reviewer 1 Report

 In this manuscript entitled "Chromium nanoparticles together with a switch away from high-fat/low-fiber dietary habits enhance the pro-healthy regulation of liver lipid metabolism and inflammation in obese rats " authors Bartosz et al., postulated that metabolic disturbances in the liver tissue associated with chronic intake of an obesogenic diet could be subsequently alleviated through dietary supplementation with various forms of chromium or switching to a low-fat diet.  

     My specific concerns/comments are below:

1.     Why authors didn’t perform pair feeding? Since it is a diet-induced obesity study, the authors must ensure that the food consumed by the control and experimental groups is the same. 

2.     Why didn’t the authors perform any toxicity study of these nanoparticles? Is there any previous report showing liver pathology from these supplements? If so, please cite it in the discussion part. 

3.     How did the authors choose the dose of Cr-Pic and Cr-NP as 0.3 mg/kg body weight? 

Author Response

Reviewer 1

In this manuscript entitled "Chromium nanoparticles together with a switch away from high-fat/low-fiber dietary habits enhance the pro-healthy regulation of liver lipid metabolism and inflammation in obese rats " authors Bartosz et al., postulated that metabolic disturbances in the liver tissue associated with chronic intake of an obesogenic diet could be subsequently alleviated through dietary supplementation with various forms of chromium or switching to a low-fat diet.  

     My specific concerns/comments are below:

  1. Why authors didn’t perform pair feeding? Since it is a diet-induced obesity study, the authors must ensure that the food consumed by the control and experimental groups is the same. 

Response: We strongly agree that the paired feeding methodology is attributed by line of advantages, e.g. a distinct advantage over ad libitum method as regards to the adaptability of the results to statistical treatment. But, there are some disadvantages, for instance i) the case that the faster-growing animal is penalized because of restricted feeding, ii) the larger animal must be using a larger proportion of feed for maintenance and less remains for growth promotion, iii) the frequent effect of a nutritionally deficient ration is to decrease feed consumption, iiii) by limiting feed intake, the full effect of the better ration cannot express itself, iiiii) the method is not suitable for finding out how much superior one ration is to another for growth. Our experimental schema and statistical approach can also be justified by the similar previous and preliminary experiments that were the basement for reviewed and accepted scientific project. The main intention was to mimic in in vivo study the different human consumer dietary activity but without any food/feed restriction. Please accept our explanation.

  1. Why didn’t the authors perform any toxicity study of these nanoparticles? Is there any previous report showing liver pathology from these supplements? If so, please cite it in the discussion part. 

Response: According to the literature and  own previous preliminary studies on chromium nanoparticles as well as other nano-metals (Cu-NPs) the applied dosages should have had no toxic effect. But, as in the case of copper nanoparticles, the Cr-NPs had some negative effects on liver functions and histological parameters. In the Introduction section we have provided information, including hepatic parameters, about our previous studies on broiler fed dietary chromium nanoparticles: “In a recent experiment on broiler chickens [16], the addition of Cr at levels of 3 and 6 mg/kg of diet (irrespective of the form used—Cr-Pic or Cr-NPs) reduced abdominal fat and stimulated the blood antioxidant defence system but disturbed liver function and caused histopathological changes in the pancreas and liver. In chickens fed diets with Cr-NPs, a significant degree of hyperaemia of the hepatic and pancreatic tissue, as well as extensive foci of fatty degeneration in these organs, were noted [16]. StÄ™pniowska et al. [17] found that the addition of Cr at 3 mg/kg to the diet of broilers, irrespective of the form used—Cr-Pic or Cr-NPs—affected the hormone levels of carbohydrate metabolism (in-creasing insulin levels and reducing glucagon levels) and adversely affected the antioxidant status of the liver and breast muscle of birds.”

Ognik, K.; Drażbo, A.; Stępniowska, A.; Kozłowski, K.; Listos, P.; Jankowski, J. The effect of chromium nanoparticles and chromium picolinate in broiler chicken diet on the performance, redox status and tissue histology. Anim. Feed Sci. Technol. 2020, 259, 114326. doi:10.1016/j.anifeedsci.2019.114326

Stępniowska, A.; Drażbo, A.; Kozłowski, K.; Ognik, K.; Jankowski, J. The effect of chromium nanoparticles and chromium picolinate in the diet of chickens on levels of selected hormones and tissue antioxidant status. Animals 2020, 10, 45. doi:10.3390/ani10010045

Our recent additional findings regarding dietary chromium nanoparticles effects in rat’s tissues, including liver, have been added to the discussion section as suggested.

“The previous own experiments showed negative oxidation consequences of dietary Cr-NPs (0.3 mg kg BW) in different organs of rat, i.e. in the small intestine, liver, brain, heart, thoracic aorta [20,21,22]. That dietary pharmacologically relevant dose of chromium-nanoparticles also caused some disturbances in minerals blood distribution [23], however without excessive accumulation of chromium in rat’s tissues, including liver [24].”

  1. Dworzański, W.; Cholewińska, E.; Fotschki, B.; Juśkiewicz, J.; Listos, P.; Ognik, K. Assessment of DNA methylation and oxidative changes in the heart and brain of rats receiving a high-fat diet supplemented with various forms of chromium. Animals 2020, 10, 1470. doi:10.3390/ani10091470
  2. Majewski, M.; Gromadziński, L.; Cholewińska, E.; Ognik, K.; Fotschki, B.; Juśkiewicz, J. Dietary effects of chromium picolinate and chromium nanoparticles in Wistar rats fed with a high-fat, low-fiber diet: the role of fat normalization. Nutrients 2022, 14, 5138. doi:10.3390/nu14235138
  3. Dworzański, W.; Cholewińska, E.; Fotschki, B.; Juśkiewicz, J.; Ognik, K. Oxidative, epigenetic changes and fermentation processes in the intestine of rats fed high-fat diets supplemented with various chromium forms. Sci. Rep. 2022, 12, 9817. doi:10.1038/s41598-022-13328-5
  4. Stępniowska, A.; Tutaj, K.; Juśkiewicz, J.; Ognik, K. Effects of a high-fat diet and chromium on hormones level and Cr retention in rats. J. Endocrinol. Invest. 2022, 45, 527-535. doi:10.1007/s40618-021-01677-3
  5. Stępniowska, A.; Juśkiewicz, J.; Tutaj, K.; Fotschki, J.; Fotschki, B.; Ognik, K. Effect of chromium picolinate and chromium nanoparticles added to low- and high-fat diets on chromium biodistribution and the blood level of selected minerals in rats. Pol. J. Food Nutr. Sci. 2022, 72(3), 229-238. doi:10.31883/pjfns/151750
  6. How did the authors choose the dose of Cr-Pic and Cr-NP as 0.3 mg/kg body weight? 

Response: The dosage 0.3 mg Cr(III) per kg BW challenged to rats was taken according to a Tolerable Daily Intake (TDI) from the Scientific Opinion on Dietary Reference Values for chromium (EFSA Panel on Dietetic Products, Nutrition and Allergies.  EFSA Journal 2014;12(10):3845) with no observed adverse effects. There is no doubt that some of the consumers are taken mineral supplements carelessly. Supplements containing only chromium are freely available, and they commonly provide 200 µg to 500 µg chromium, although some contain up to 1000 µg (Costello et al., 2019. Chromium supplements in health and disease. In: Vincent JB, ed. The Nutritional Biochemistry of Chromium (III). Cambridge, MA: Elsevier; 2019:219-59). This seems to be an easy way to consume high doses of Cr(III). In fact, in the present experiment the applied dose is quite high, both in a rat diet and when calculated from animal dose to the human one. Therefore, the applied dose should be regarded as the pharmacologically relevant dose of Cr and the appropriate information have been put in the text.

I hope you will find the revisions satisfactory, and I and all co-authors look forward to hearing your response to the revised manuscript.

Sincerely,

Jerzy Juśkiewicz and co-authors

Reviewer 2 Report

Thank you for submitting this article. While it was not the easiest article to read, it was interesting.

Some things are to be improved :

1. At line 91: This is the first time in the text where we see the abbreviations M, F, MP, FP, MN, FN. Although we later see the definitions in the figure description, this should be defined right there in the text, even more so since ''Methods'' is at the end of the paper. This should help readers have a clear vision of what each treatment is. I spent more time than was necessary going back and forth between this part and the figure description just to know what the treatments were. Must be corrected.

2. Between line 91-114, there is a lot of informations describing absolutely everything the authors found, even if it isn't really relevant to what the authors try to say. Description of literally every statistical test results makes this part very heavy and confusing, making it less clear as to what is really relevant. This absolutely must be improved and results going in the way of Cr nanoparticles combined with a switch away from high fat diet must be highlighted so it is better understood. 

3. In the same way, at line 99 it is written ''the daily BW gain of group FN differed significantly from that of all other groups''. Is it higher? lower? Yes, i can look in the table, but why explain every single other results and not explain this one in particular?

4. Possible error at line 116 ''...but 111case for the M and MP groups...'' What is 111case?

5. The description of every treatment is in every figure/table description. This is not necessary and makes the text heavy. Furthermore, it would be better to explain completely the treatments. For example, if a reader didn't read the method or result texts, how could he differentiate between C and M? One is '' control, fed a C-diet '' and the other is ''fed a C-diet''. This should absolutely be corrected to indicate that M went through the Fat diet first and then was reared on C diet.

6. One part that is relatively confusing is, is the ''C'' diet a special low fat diet or a control diet ? In Methods, this has to be clarified, indicating clearly the % of Fat, carbohydrates, and proteins for each diets.

Author Response

Reviewer 2

Thank you for submitting this article. While it was not the easiest article to read, it was interesting.

Response: We thank the Reviewer for his comments.

Some things are to be improved :

  1. At line 91: This is the first time in the text where we see the abbreviations M, F, MP, FP, MN, FN. Although we later see the definitions in the figure description, this should be defined right there in the text, even more so since ''Methods'' is at the end of the paper. This should help readers have a clear vision of what each treatment is. I spent more time than was necessary going back and forth between this part and the figure description just to know what the treatments were. Must be corrected.

Response: You are right. It has been corrected as suggested. It has been added the following text: “The feeding period consisted of an initial 9-wk and an experimental 9-wk period. During the initial period, C rats were fed a standard low-fat diet (diet C), while the remaining groups (M, F, MP, FP, MN, FN) were subjected to an obesogenic high-fat diet (diet F). The dietary treatments used in the experimental period: Group C, control fed a C-diet; M, fed a C-diet; F, fed an F-diet; MP, fed a C-diet with Cr-Pic supplementation; FP, fed an F-diet with Cr-Pic; MN, fed a C-diet with Cr-NP; FN, fed a F-diet with Cr-NP. Both forms of Cr were added in a dose of 0.3 mg/kg body weight (BW).”

  1. Between line 91-114, there is a lot of information describing absolutely everything the authors found, even if it isn't really relevant to what the authors try to say. Description of literally every statistical test results makes this part very heavy and confusing, making it less clear as to what is really relevant. This absolutely must be improved and results going in the way of Cr nanoparticles combined with a switch away from high fat diet must be highlighted so it is better understood.

Response: The Results section has been rewritten according to Reviewer’s suggestion. Taking into account the main hypothesis, the unnecessary sentences have been omitted.

  1. In the same way, at line 99 it is written ''the daily BW gain of group FN differed significantly from that of all other groups''. Is it higher? lower? Yes, i can look in the table, but why explain every single other results and not explain this one in particular?

Response: The sentence has been corrected.

  1. Possible error at line 116 ''...but 111case for the M and MP groups...'' What is 111case?

Response: The error at line 116 has been corrected.

  1. The description of every treatment is in every figure/table description. This is not necessary and makes the text heavy. Furthermore, it would be better to explain completely the treatments. For example, if a reader didn't read the method or result texts, how could he differentiate between C and M? One is '' control, fed a C-diet '' and the other is ''fed a C-diet''. This should absolutely be corrected to indicate that M went through the Fat diet first and then was reared on C diet.

Response: The description has been redrafted and it has been provide in full under Table 1. To improve description in remaining tables and figures an appropriate reference to Table 1 has been added instead of full text.

*The feeding period consisted of an initial 9-wk and an experimental 9-wk period. During the initial period, group C rats were fed a diet with 8% rapeseed oil and 64% maize starch (low-fat diet C), while the remaining groups were subjected to an obesogenic diet with 8% rapeseed oil, 17% lard, and 52% maize starch (high-fat diet F). The dietary treatments used in the experiment were as follows: Group C, control, fed a C-diet during the initial and the experimental periods; M, fed an F-diet during the initial period and a C-diet during the experimental period; F, fed an F-diet during the initial and the experimental periods; MP, fed an F-diet during the initial period and a C-diet with supplementation of chromium picolinate (Cr-Pic; 0.3 mg/kg BW) during the experimental period; FP, fed an F-diet during the initial period and an F-diet with supplementation of Cr-Pic (0.3 mg/kg BW) during the experimental period; MN, fed an F-diet during the initial period and a C-diet with supplementation of chromium nanoparticles (Cr-NP; 0.3 mg/kg BW) during the experimental period; FN, fed an F-diet during the initial period and an F-diet with supplementation of Cr-NP (0.3 mg/kg BW) during the experimental period.

  1. One part that is relatively confusing is, is the ''C'' diet a special low fat diet or a control diet ? In Methods, this has to be clarified, indicating clearly the % of Fat, carbohydrates, and proteins for each diets.

Response: Thank you for that remark. The C diet was a low-fat diet (a standard semi-purified diet for rats prepared in our lab). In the “Materials and Methods” section the calculated nutritional values (kcal% for protein, fat and carbohydrates) for experimental diets have been provided. Same in Table 5.

I hope you will find the revisions satisfactory, and I and all co-authors look forward to hearing your response to the revised manuscript.

Sincerely,

Jerzy Juśkiewicz and co-authors